# Evaluating Approximations and Heuristic Measures of Integrated Information

**DOI:** 10.3390/e21050525

**Published:** 2019-05-24

**Authors:** André Sevenius Nilsen, Bjørn Erik Juel, William Marshall

**Affiliations:** 1Brain Signalling Group, Department of Physiology, Institute of Basic Medicine, University of Oslo, Sognsvannsveien 9, 0315 Oslo, Norway; 2Department of Psychiatry, University of Wisconsin, Madison, WI 53719, USA; 3Department of Mathematics and Statistics, Brock University, St. Catharines, ON L2S 3A1, Canada

**Keywords:** integrated information theory, differentiation, integration, complexity, consciousness, computational, IIT, Phi

## Abstract

Integrated information theory (IIT) proposes a measure of integrated information, termed Phi (Φ), to capture the level of consciousness of a physical system in a given state. Unfortunately, calculating Φ itself is currently possible only for very small model systems and far from computable for the kinds of system typically associated with consciousness (brains). Here, we considered several proposed heuristic measures and computational approximations, some of which can be applied to larger systems, and tested if they correlate well with Φ. While these measures and approximations capture intuitions underlying IIT and some have had success in practical applications, it has not been shown that they actually quantify the type of integrated information specified by the latest version of IIT and, thus, whether they can be used to test the theory. In this study, we evaluated these approximations and heuristic measures considering how well they estimated the Φ values of model systems and not on the basis of practical or clinical considerations. To do this, we simulated networks consisting of 3–6 binary linear threshold nodes randomly connected with excitatory and inhibitory connections. For each system, we then constructed the system’s state transition probability matrix (TPM) and generated observed data over time from all possible initial conditions. We then calculated Φ, approximations to Φ, and measures based on state differentiation, coalition entropy, state uniqueness, and integrated information. Our findings suggest that Φ can be approximated closely in small binary systems by using one or more of the readily available approximations (*r* > 0.95) but without major reductions in computational demands. Furthermore, the maximum value of Φ across states (a state-independent quantity) correlated strongly with measures of signal complexity (LZ, *r*_s_ = 0.722), decoder-based integrated information (Φ*, *r*_s_ = 0.816), and state differentiation (D1, *r*_s_ = 0.827). These measures could allow for the efficient estimation of a system’s capacity for high Φ or function as accurate predictors of low- (but not high-)Φ systems. While it is uncertain whether the results extend to larger systems or systems with other dynamics, we stress the importance that measures aimed at being practical alternatives to Φ be, at a minimum, rigorously tested in an environment where the ground truth can be established.

## 1. Introduction

The nature of consciousness, defined as a subjective experience, has been a philosophical topic for centuries but has only recently become incorporated into mainstream neuroscience [1]. However, as consciousness is a subjective phenomenon, and thus not directly measurable, it must be operationalized to allow for empirical investigation of its nature and underlying mechanisms [2]. In other words, the scientific study of consciousness requires an objective measure. One such measure has been developed within the framework of the integrated information theory (IIT), introduced and elaborated by Giulio Tononi and colleagues [3,4,5]. The theory has attracted much interest because of its axiomatic quantitative approach towards illuminating fundamental aspects of consciousness. The theory proposes that consciousness is identical to a particular type of integrated information (Phi; Φ) which is defined and quantified within the theory as a measure of a system’s informational irreducibility, or how much information a system in a definite state specifies about its own past and future above and beyond how much such information is specified by its parts.

A major practical limitation of IIT is the computational cost of calculating Φ, which, according to the current formulation (version 3.0 [5]; here referred to as Φ_3.0_, implemented through PyPhi [6]), grows as O(*n*53*^n^*) [6] for binary systems where *n* is the number of elements in the system. In addition, computing Φ_3.0_ requires full knowledge of a system’s transition probabilities (the probability of the system transitioning from any state to any other state). Taken together, these knowledge and computational requirements place strong constraints on both the system size and the level of possible precision for which Φ_3.0_ can be calculated. Therefore, the exact value of Φ_3.0_ is intractable for most biological or artificial systems of interest. Currently, the largest systems being investigated are in the order of 20–30 binary elements [7,8], with a practical limit of ~10–12 elements, unless special assumptions are made about the system under investigation (e.g., see [9]).

As Φ_3.0_ quickly becomes computationally intractable as a function of network size, one approach is to implement approximations (computational shortcuts) within the framework of IIT_3.0_ that reduce the computational cost [6]. Another approach is to use heuristic measures that capture central intuitions of IIT such as information differentiation and integration via more tractable methods [10,11,12,13,14,15]. While many heuristics have been applied to electrophysiological data (e.g., [10,13,14,16,17,18]), simulated time series of continuous variables (e.g., [11,19]), and discrete variables (e.g., [15,20]), only [15] have tested a few approximations and heuristics with respect to Φ_3.0_ in evolved logic-gate-based animats. Notably, a study [19] compared the behavior of several heuristic measures developed for time-series data; however, the authors were interested in the consistency among the methods, rather than in a comparison with Φ_3.0_.

The lack of direct comparisons with Φ_3.0_ is a gap in the current literature of integrated information methods. If an approximation or heuristic is to be used in an attempt to falsify IIT, then the results are only valid to the extent that the measure accurately estimates Φ_3.0_ (similarly, for evidence in favor of IIT). It is not possible to validate the proposed measures in the networks of interest (due to the computational considerations outlined above); however, we can validate the measures in smaller systems where Φ_3.0_ can be calculated directly. We claim that correspondence in smaller systems is a necessary condition for any measure used to evaluate IIT. Therefore, by using deterministic, isolated, discrete networks of binary logic gates of similar type as those employed in IIT_3.0_ [5], this paper aims to evaluate the accuracy relative to Φ_3.0_ of (1) approximations that speed up parts of Φ_3.0_ calculations and (2) heuristic measures of integrated information. 

## 2. Materials and Methods 

### 2.1. Networks

We randomly generated networks consisting of n ∈ {3, ..., 6} binary linear threshold nodes (state S ∈ {0,1}), with fixed threshold (θ = 1) and weighted connections between nodes (W_ij_ ∈ {1,0,−1}, for i,j = 1, ..., *n*). There were no self-connections (W_ii_ = 0). Connections were generated as follows: First, for all i ≠ j, we set W_ij_ = 1 with a probability p ∈ {0.2, 0.3, …, 1.0}, a parameter that was fixed for each network. Second, we changed the sign of non-zero connections to W_ij_ = −1 with probability q ∈ {0.0, 0.1, …, 0.8}; this parameter was also fixed for each network. The remaining weights were kept at W_ij_ = 0, i.e., no connection. Altogether, the connections were independent, with Pr(W_ij_ = 1) = p(1 − q) and Pr(W_ij_ = −1) = pq, and Pr(W_ij_ = 0) = 1 − p. To avoid duplicate network architectures, all networks were checked for uniqueness up to an isomorphism of nodes, i.e., two networks were considered equal if they could be mapped to each other by a relabeling of nodes (using a brute force algorithm). The networks were isolated (no external inputs or modulators). In sum, we generated networks with nodes that could take one of two states (S_t_ = 0, 1) and would be activated (S_t+1_ = 1) if the weighted sum of the inputs to the node was equal to or larger than its threshold (θ = 1). If a node was activated, it would then output to other nodes according to its outgoing connection weights. Importantly, this allowed for networks with excitatory (W_ij_ = 1), inhibitory (W_ij_ = −1), and no (W_ij_ = 0) connection between any given pair of nodes (see Figure 1a).

To investigate various measures and approximations, we needed functional information about the networks in the form of a probabilistic description of the transitions from any given state to any other state, i.e., a transition probability matrix (TPM). For each network, a TPM was constructed based on the node mechanism (linear threshold with θ = 1) and the connection weights W_ij_. As the generated networks were deterministic, the TPM contained only a single ‘1’ in each row representing the next state of the network. 

From the TPM, given an initial condition, we were able to generate “observed” time-series data for each network. From a given initial condition, a network may only explore part of its state space before reaching an attracting fixed point or periodic sequence. While generating the observed data, we periodically perturbed the network into a new state, ensuring that our data fully explored the state space of the network and that the results were not dependent on our choice of initial condition. This procedure resembles the perturbations applied by transcranial magnetic stimulation (TMS)during empirical studies of consciousness [14]. The generated time-series data consisted of 2*^n^* epochs, where one epoch was generated by initializing/perturbing a network to an initial state and then was simulated for a total of α(*n*)(2*^n^* + 1) timesteps. The function α(*n*) ensured parity of bits between the generated time series for networks of different sizes (see Section A.1). This perturbation and simulation process was repeated for all possible network states (2*^n^*) sequentially, with each epoch appended to the last preceding epoch. The resulting simulated time series (sequence of epochs) produced an α(*n*)(2*^n^* + 1)2*^n^-*by-*n* matrix where each of the *n* columns reflected the state of a single node over time, and each row reflected the current state of each network node (0/1) at a given time. In sum, we derived a TPM from the mechanism and connectivity profile of individual nodes and then, using the TPM and perturbations, generated a time series of observed data that explored the entire state space of the network (see Figure 1b,c).

### 2.2. Integrated Information

For the networks defined above, we calculated Φ_3.0_ as implemented through PyPhi v1.0 [6]. Here, we just give a brief summary of how Φ_3.0_ was defined and calculated, but see reference [5] for a more detailed account. Generally, IIT proposes that a physical system’s degree of consciousness is identical to its level of state-dependent causal irreducibility (Φ^max^), i.e., the amount of information of a system in a specific state above and beyond the information of the system’s parts. 

The calculation of Φ_3.0_ began with “mechanism-level” computations. For a given *candidate system* (subset of a network) in a state, we identified all possible *mechanisms* (subsets of system nodes in a state that irreducibly constrained the past and future state of the system). For each mechanism, we considered all possible purviews (subsets of nodes) that the mechanism constrained. For a given mechanism–purview combination, we found its *cause–effect repertoire* (CER; a probability distribution specifying how the mechanism causally constrained the past and future states of the purview). To find the irreducibility of the CER, the connections between all permissible bipartitions of elements in the purview and the mechanism were *cut* (see [6]); the bipartition producing the least difference is called the *minimum information partition* (MIP). Irreducibility, or integrated information, φ, is quantified by the earth mover’s distance (EMD) between the CER of the uncut mechanism and the CER of the mechanism partitioned by the MIP. A mechanism, together with the purview over which its CER is maximally irreducible and the associated φ value, specifies a *concept*, which expresses the causal role played by the mechanism within the system. The set of all concepts is called the *cause–effect structure* of the candidate system. 

Once all irreducible mechanisms of a candidate system were found, a similar set of operations was done at the “system level” to understand whether the set of mechanisms specified by the system were reducible to the mechanisms specified by its parts. The irreducibility of the candidate system was quantified by its conceptual integrated information, Φ. This process was repeated for all candidate systems, and the candidate system that was maximally irreducible among all candidate systems was termed a *major complex* (MC). According to IIT then, the MC was the substrate that specified a particular conscious experience for the (physical) system in a state, and Φ_3.0_ quantified the irreducibility of the cause–effect structure it specified in that state. As such, Φ_3.0_ was calculated for every reachable state of the system, i.e., state-dependently. 

As many of the heuristics and approximations outlined below are state-independent, there is no direct comparison to the state-dependent Φ_3.0_. To facilitate comparisons with these measures, we further computed a state-independent quantity, $\Phi_{3.0}^{peak}$, as the maximum value of Φ_3.0_ across all states of the network. The quantity $\Phi_{3.0}^{peak}$ can be thought of as a measure of a network capacity for consciousness, rather than its currently realized level of consciousness. Alternatively, we could also compute the mean value of Φ_3.0_, which has some relation to the state-dependent value of Φ_3.0_ under certain regularity conditions [15], but the results were similar (see Figure 5d).

### 2.3. Approximations and Heuristics

To speed up the calculation of Φ_3.0_, one can implement several shortcuts or approximations based on assumptions about the system under consideration. Here, we aimed to test six specific approximations; three approximations that are already implemented in the toolbox for calculating Φ_3.0_ (PyPhi; [6]) that reduce the complexity of evaluating information lost during partitioning of a network; two shortcuts based on estimating the elements included in the MC rather than explicitly testing every candidate subsystem; and one estimation of a system’s $\Phi_{3.0}^{peak}$ from the Φ of a few states, rather than taking the maximum over all possible states. All approximations were likely to compare well against Φ_3.0_, but were unlikely to yield significant savings in computational demand. 

Another approach is to use heuristics that capture aspects of Φ_3.0_. These heuristics can be separated into two classes: those that require the full TPM and discrete dynamics (heuristics on discrete networks requiring perturbational data) and those that require time-series data (heuristics from observed data). While these measures may reduce the computational demands, the heuristics based on discrete dynamics still require full structural and functional knowledge of the system, which reduces their applicability. On the other hand, measures based on observed data significantly broaden the potential applicability at the cost of estimating the underlying causal structure by using the observed time series.

All approximations and heuristics that were tested are listed in Table 1, together with an identifier (from “A” to “N”) that will be used in the text for ease of reading, as well as a reference and brief description.

#### 2.3.1. Approximations to Φ_3.0_

We calculated several approximations to Φ_3.0_. (A) The cut-one approximation (CO) reduced the number of partitions considered when searching for the MIP. The approximation assumes that the MIP is achieved by cutting only a single node out of the candidate system; (B) the no-new-concepts approximation (NN) eliminates the need to rebuild the entire cause–effect structure for every partition under the assumption that when a partition is made it does not give rise to new concepts. Thus, one only needs to check for changes to existing mechanisms, rather than reevaluating the entire powerset of potential mechanisms.

We also tested two approximations based on estimates of which nodes are included in the MC. These approximations assumed the MC consisted of either (C) all the nodes in the system taken as a whole (whole system; WS), or (D) the subsystem of the network where all nodes with no recursive connectivity (no input and/or output connections) or an unreachable state (nodes that were always “on” or always “off”, such as a node with only inhibitory inputs) had been removed, iteratively (iterative cut; IC). Note that by unreachable, we mean there was no state of the network that would lead to a particular node being “on” (or “off”) in the next time step. This does not mean that we could not use an external perturbation to set the node into any state (which we did when generating the observed data). In IIT_3.0_, such a node (either with no inputs, no outputs, or an unreachable state) can be partitioned without loss, leading to Φ_3.0_ = 0. Simply excluding these nodes from the MC is not an approximation but a computational shortcut, as they will necessarily be outside the MC. However, the approximation consisted in assuming that the remaining set of recursively connected nodes was the MC.

As with Φ_3.0_, these measures were calculated in a state-dependent and state-independent manner. Finally, we tested (E) if the state-independent $\Phi_{3.0}^{peak}$ could be estimated by randomly sampling the state-dependent Φ_3.0_, termed here “Est.*n*$\Phi_{3.0}^{peak}$”, where *n* refers to the number of samples (*n* = 1,2, ..., 15).

#### 2.3.2. Heuristics on Discrete Networks

To estimate Φ_3.0_, we investigated several heuristic measures defined for discrete networks. While the latest iteration of IIT takes steps to make the mathematical formalism more in tune with the intended interpretation of its axioms and postulates, IIT_3.0_ is more computationally intractable than previous versions (see S1 of [5]). To compare the results of the two newest versions of the theory, we tested (F) Φ based on IIT_2.0_, Φ_2.0_ [3], and (G) Φ_2.0_ incorporating minimization over both cause–effect and not only cause, Φ_2.5_ [12]. These measures are, however, still limited by the exponential growth in computational time and are included here because IIT_2.0_ was used as inspiration for other measures, and their validity depends on the correspondence between IIT_2.0_ and IIT_3.0_.

As Φ_3.0_ is sensitive to a large state repertoire, i.e., divergent and convergent behavior-weakening cause/effect constraints (assuming irreducibility), we also included two measures that captured the dynamical differentiation of states in the system; (H) The number of reachable states, D1, quantifying the system’s available repertoire of states, and (I) cumulative variance of system elements, D2, indicating the degree of difference between system states [15]. For D1, we calculated the number of states that were reachable, i.e., states that had a valid precursor state. Accordingly, D1 was inversely related to a system’s degeneracy of state transitions. D2 calculated the cumulative variance of activity in each system node given the maximum entropy distribution of initial conditions. As such, D2 reflected how different the system’s reachable states were from each other. See [15] for a more thorough account.

Both Φ_2.0_ and Φ_2.5_ were calculated in a state-dependent and in a state-independent manner ($\Phi_{2.0}^{peak}$/$\Phi_{2.5}^{peak}$), while both D1 and D2 were only defined state-independently. All the heuristics on discrete systems were calculated using the system TPM. As such, while these measures were faster to calculate and flexible in terms of network size, they still required full knowledge of the functional dynamics of the system (i.e., the full TPM).

#### 2.3.3. Heuristics from Observed Data

To alleviate the full knowledge requirement, we considered heuristic measures that are defined for observed (time-series) data. Given their relative success in distinguishing conscious from unconscious states in experiments and clinical populations [13,22,23] and their apparent similarity to central IIT intuitions, we focused on measures of signal diversity. There are many candidates to choose from, but here, we included (J) coalition entropy (*S*), measured by the entropy of the observed state distribution indicating a system’s average diversity of visited states [22], and (K) signal complexity measured by algorithmic compressibility through Lempel-Ziv compression (LZ), indicating the degree of order or patterns in the observed state sequences of a system [22]. Both entropy and complexity measures have been used in EEG to distinguish between states of consciousness [13,24]. 

In addition, several measures have been developed that share many of IITs underlying intuitions, such as capturing integrated information of a system above and beyond its parts while staying computationally tractable [10,11,19,21,25]. Although these measures can be applied to continuous data in the time domain such as EEG, here, we focused on a selection of these measures that can be applied to discrete, binary data. Specifically, we tested: (L) decoder-based integrated information (Φ*) based on IIT_2.0_ [21], (M) integrated stochastic interaction (SI) based on IIT_2.0_ [11], and (N) mutual information (MI) based on IIT_1.0_ [21]. The integrated information measures were implemented using the “Practical PHI toolbox for integrated information analysis” [26] with the discrete forms of the formulae, employing a MIP exhaustive search with a bipartition scheme (powerset; 2*^n^*^−1^−1) and a normalization factor according to IIT_2.0_ [3]. All heuristics were calculated in a state-independent manner, using the time-series data generated for the whole network (no searching through subsystems).

### 2.4. Analysis

Comparisons between Φ_3.0_ and approximate measures (CO, NN, WS, IC) were analyzed using Pearson correlations (*r*) and separate ordinary least-squares linear regression models as the approximations were expected to be closely related to Φ_3.0_. Statistics of linear fits are reported. For comparisons between Φ_3.0_ and all other measures we used Spearman’s correlation (*r*_s_) to investigate the monotonicity of the relationship, as a linear relationship was not necessarily expected. All state-dependent measures were compared to Φ_3.0_, while all state-independent measures were compared to $\Phi_{3.0}^{peak}$. Metrics of significance (*p* values) are not reported because of our large sample size; for our sample (n > 1981), correlations as small as |*r*| = 0.044 were statistically significant at the 0.05 level, but such small correlations were not meaningful in the context of the study. As we focused on high correspondence, we instead report correlations as weak, 0.5 < *r* < 0.7, medium 0.7 < *r* < 0.8, strong 0.8 < *r* < 0.9, and very strong, *r* > 0.9 (for both *r* and *r*_s_). 

### 2.5. Setup

Calculation of measurements was performed in Python (v3.6) with PyPhi (v1.0) [6] for Φ_3.0_, CO, NN, WS, and IC; Matlab (v2016b) with “Practical PHI toolbox for integrated information analysis” (v1.0) [26] for Φ*, SI, MI; custom code in Python (v3.6) for Φ_2.0_, Φ_2.5_, D1, D2; and Python (v3.6) with scripts from [13] for LZ, and *S*. Statistics were done with custom code in Python (v3.6) and Statsmodels (v.0.8.0). Everything else was done with custom code in Python (v3.6), Numpy (v1.13.1), SciPy (v0.19.1), and Pandas (v0.20.3).

## 3. Results

We analyzed 2032 randomly generated networks, with 131 three-node, 675 four-node, 866 five-node, and 360 six-node networks. In total, 61,224 states were analyzed. Note that the heuristic measures were only analyzed in 309 of the six-node networks due to time constraints. See Table 2 for an overview of the main results and Figure 2 for four example networks.

### 3.1. Descriptive Statistics

Mean and variance of Φ_3.0_ grew as a function of network elements (*n* = 3: M = 0.015 ± 0.121SD to *n* = 6: M = 0.386 ± 0.487SD). As the systems increased in size, the fraction of $\Phi_{3.0}^{peak}$ = 0 networks (indicating a completely reducible system, e.g., a feedforward network) decreased. We also monitored a class of networks with $\Phi_{3.0}^{peak}$ = 1, as this typically indicated that the MC was a stereotyped unidirectional “loop”. The fraction of these stereotyped networks stayed relatively stable as *n* increased, while the fraction of networks with $\Phi_{3.0}^{peak}$ > 1 increased. See Figure 3.

### 3.2. Approximations

Both the no-new-concepts (NN) and the cut-one (CO) approximations were nearly perfectly correlated with state-dependent (S.D.) Φ_3.0_ and state-independent (S.I.) $\Phi_{3.0}^{peak}$ (*r* > 0.996). Regression analysis showed that both no-new-concepts and cut-one approximations were strong linear predictors; S.I.: R^2^ > 0.999, *NN*$\Phi_{3.0}^{peak}$ = 0.00 + 1.00$\Phi_{3.0}^{peak}$. S.D.: R^2^ > 0.999, *NN*Φ_3.0_ = 1.00Φ_3.0_, and, S.I.: R^2^ = 0.994, *CO*$\Phi_{3.0}^{peak}$ = 0.00 + 1.04$\Phi_{3.0}^{peak}$). S.D.: R^2^ = 0.995, *CO*Φ_3.0_ = 1.02Φ_3.0_, respectively. See Figure 4a,b.

In regard to estimating $\Phi_{3.0}^{peak}$, we took samples from *n* = 1, 2, ..., 15 states with results ranging from weak correlation (*n* = 1, *r* = 0.688) to strong correlation (*n* = 15, *r* = 0.893) as the number of samples increased (for *n* = 5; R^2^ = 0.738, *SS*$\Phi_{3.0}^{peak}$ = 0.097 + 0.262Φ_3.0_). This was in accordance with a very strong correlation between $\Phi_{3.0}^{peak}$ and $\Phi_{3.0}^{mean}$ (R^2^ > 0.846, $\Phi_{3.0}^{mean}$ = 0.087 + 0.274$\Phi_{3.0}^{peak}$). These strong correlations suggest that a network with a high value of $\Phi_{3.0}^{peak}$ typically has several states with high Φ_3.0_ values, not just a single state of high Φ_3.0_. See Figure 5g,h.

Finally, we tested whether the estimated MCs could predict Φ_3.0_. WS$\Phi_{3.0}^{peak}$ was very strongly correlated with S.I.$\Phi_{3.0}^{peak}$ (R^2^ > 0.954, with *WS*$\Phi_{3.0}^{peak}$ = −0.255 + 0.986$\Phi_{3.0}^{peak}$) and with S.D. Φ_3.0_ (R^2^ > 0.876, with *WS*Φ_3.0_ = -0.163 + 0.899Φ_3.0_). ICΦ_3.0_ was very strongly correlated with S.I.$\Phi_{3.0}^{peak}$ (R^2^ > 0.974, with *IC*$\Phi_{3.0}^{peak}$ = −0.167 + 0.995$\Phi_{3.0}^{peak}$) and very strongly correlated with Φ_3.0_ (R^2^ > 0.912, with *IC*Φ_3.0_ = −0.119 + 0.927Φ_3.0_). See Figure 4e–h.

Together, these results suggest that the tested approximations can be used as strong predictors of Φ; however, these approximations still require knowledge of the systems TPM, and their computational cost grows exponentially, leading to only a marginal increase in the size of networks that can be analyzed (see Section A.4).

### 3.3. Heuristics

The state differentiation measures D1 and D2 showed strong (*r*_s_ = 0.827) and medium (*r*_s_ = 0.718) rank order correlations with S.I.$\Phi_{3.0}^{peak}$, respectively (see Figure 5e,f).

S.D. Φ_2.0_ and Φ_2.5_ were weakly or less correlated with Φ_3.0_ (*r*_s_ = 0.622 and *r*_s_ = 0.473, respectively), while S.I. variants of Φ_2.0_ and Φ_2.5_ were strongly rank-order correlated with $\Phi_{3.0}^{peak}$(*r*_s_ = 0.838 and *r*_s_ = 0.832, respectively) (Figure 5a,b).

The state-independent heuristic LZ and *S* were medium correlated with $\Phi_{3.0}^{peak}$ (0.71 < *r*_s_ < 0.72) (Figure 5c, only LZ shown). The state-independent measures SI and MI were weakly or less correlated with $\Phi_{3.0}^{peak}$ (*r*_s_ < 0.54), while Φ* was strongly rank-order correlated with $\Phi_{3.0}^{peak}$, (*r*_s_ = 0.82) (Figure 5d, only Φ* shown). For Φ*, the results showed two clusters of values, one seemingly linearly related to $\Phi_{3.0}^{peak}$, and one non-correlated cluster consisting of low $\Phi_{3.0}^{peak}$/high Φ* outliers. A post-hoc analysis removing outliers above two standard deviations of the mean negligibly influenced the results (see Section A.2).

Together, these results suggest that the tested heuristics might be accurate predictors of $\Phi_{3.0}^{peak}$ on a group level however not necessarily for individual networks; they also drastically reduce computational demands (see Section A.4). In addition, all heuristics showed an increased variance of $\Phi_{3.0}^{peak}$ with higher values, suggesting reduced correspondence for higher values.

### 3.4. Post-hoc Tests

For all measures, removing non-integrated ($\Phi_{3.0}^{peak}$ = 0) or irreducible circular networks ($\Phi_{3.0}^{peak}$ = 1) reduced the correlational values. This was true for all heuristics, while the approximations were minimally affected. After this adjustment, S.I. D1 and Φ* were the heuristics highest correlated with $\Phi_{3.0}^{peak}$ (*r*_s_ = 0.703 and *r*_s_ = 0.698, respectively), with LZ the third (*r*_s_ = 0.616). This indicates that the results were influenced by a large cluster of non-integrated and circular networks and that the measures were sensitive to the difference between them (see Section A.3).

## 4. Discussion

We randomly generated a population of small networks (three to six nodes) with linear threshold logic and both excitatory and inhibitory connections. We evaluated several approximations and heuristic measures of integrated information based on how well they corresponded to the Φ_3.0_, according to the definition proposed by integrated information theory. The purpose of the work was to determine which methods, if any, might be used to test the theory. Since the accuracy of these methods cannot be evaluated for large networks of the size typically of interest for consciousness studies, we considered success in the current study—correspondence in small networks where Φ_3.0_ can be computed—as a minimal requirement for any such measure. In summary, we observed that the computational approximations were strong predictors (as defined in Section 2.4) of both Φ_3.0_ and $\Phi_{3.0}^{peak}$, while heuristic measures were only able to capture $\Phi_{3.0}^{peak}$. The approximation measures were still computationally intensive and required full knowledge of the systems TPM, meaning they only provided a marginal increase to the size of the systems that can be studied. Heuristic measures on the other hand, provided greater reductions in computation and knowledge requirements and can be applied to much larger systems, but only in a coarser state-independent manner.

### 4.1. Approximation Measures

The approximation measures we tested were developed by starting from the definition of Φ_3.0_ and then making assumptions to simplify the computations. Although they did not reduce computation enough to substantially increase the applicability of Φ_3.0_, their success provides a blueprint for future approximations. We discuss two aspects of Φ_3.0_ computation that should be investigated in future work: finding the MC of a network and finding the MIP of a mechanism–purview combination.

Regarding the estimates of the MC, the Φ_3.0_ value of any subsystem within a network is a lower bound on the Φ_3.0_ of the MC of that network. Moreover, the WS approximation (assuming the MC is the whole system) and the IC approximation (assuming the MC is the whole system after removing nodes without inputs or without outputs and inactive nodes) were both highly predictive of Φ_3.0_ (and of $\Phi_{3.0}^{peak}$). Estimating the MC provided computational savings by eliminating the need to compute Φ_3.0_ for all possible subsets of elements. However, the computational cost of computing Φ_3.0_ for an individual subsystem still grows exponentially with the size of the subsystem. Any MC estimate close to the full size of the network will still require substantial computation. Therefore, finding a minimal MC that still accurately estimates Φ_3.0_ would be most efficient for reducing the computational demands. While this may limit the usability of MC estimates (for highly integrated systems, the MC is more likely to be the whole system), such methods could be used to investigate questions regarding which part of a system is conscious (e.g., cortical location of consciousness [27]). 

Using the CO approximation (assuming that at the system level, the MIP results from partitioning a single node), we observed very strong correlations with Φ_3.0_ (and $\Phi_{3.0}^{peak}$). Usually, the number of partitions to check grows exponentially with the number of nodes in the system, but with the CO approximation it grew linearly, providing a substantial computational savings. Extending the CO approximation (or some variant of it, see [28,29,30]) from the system-level MIP to the mechanism-level MIPs could provide even greater computational savings. While only a single system-level MIP needs to be found to compute Φ_3.0_, a mechanism-level MIP must be found for every mechanism–purview combination (the number of which grows exponentially with the system size). 

As an aside, the IIT_3.0_ formalism only considers bipartitions of nodes when searching for the MIP, presumably on the basis that further partitioning a mechanism (or system) could cause additional information loss (and, thus, never be a *minimum* information partition). To explore this, we employed an alternative definition of the MIP requiring a search over all partitions (AP, as opposed to bipartitions) for a subset of our networks. While we observed a very high correlation between all the partitions and bipartitions schemes (S.I. $\Phi_{3.0}^{peak}$ R^2^ = 0.966; S.D. Φ_3.0_ R^2^ = 0.921; see Section A.7), the correspondence was not exact. Note that the definition of a partition used for the ‘all partitions’ option is slightly different than the definition for ‘bipartitions’, so the set of partitions in the AP option is not strictly a superset of the set of bipartitions (see PyPhi v1.0 and its documentation [6] or Section A.7 for more details). Despite this difference, we saw a very strong correlation between the methods, suggesting that different rules for permissible cuts could be considered as potential approximations.

### 4.2. Heuristic Measures

Although heuristic measures did not capture state-dependent Φ_3.0_, most were rank-correlated with state-independent $\Phi_{3.0}^{peak}$. However, all heuristic measures were negatively impacted by removing networks with $\Phi_{3.0}^{peak}$ = 0‖1, indicating that reducible ($\Phi_{3.0}^{peak}$ = 0) or circular ($\Phi_{3.0}^{peak}$ = 1) networks can confound comparisons, as a majority of networks fall in this range. The heuristics that showed the strongest correlation after removal of $\Phi_{3.0}^{peak}$ = 0‖1 networks were measures of state differentiation (D1), integrated information (Φ*), and complexity (LZ). Together, these results suggest that D1, Φ*, and, to a lesser degree, LZ could be useful heuristics for $\Phi_{3.0}^{peak}$ at the group level, although unreliable at the individual level.

The heuristic D1 measures the number of states accessible by a system [15], and the strong correlation we observed indicates that systems with a large repertoire of available states are also likely to have high $\Phi_{3.0}^{peak}$ (assuming the systems are irreducible, i.e., $\Phi_{3.0}^{peak}$ > 0). This finding is interesting because clinical results also corroborate state differentiation as a factor in unconsciousness, where it has been observed that the state repertoire of the brain is reduced during anesthesia [31]. While D1 is computationally tractable, it requires full knowledge of the system (i.e., a TPM with 2^2*n*^ bits of information), that the system is integrated, and that transitions are relatively noise-free. As such, unfortunately, D1 cannot be applied to larger artificial or biological systems of interest (such as the brain). The second measure that correlated well with $\Phi_{3.0}^{peak}$ can also be seen to quantify state differentiation to some extent. LZ is a measure of signal complexity [32], offering a concrete algorithm to quantify the number of unique patterns in a signal. While LZ has been used to differentiate conscious and unconscious states [13,33], it cannot distinguish between a noisy system and an integrated but complex one from observed data alone. Thus, some knowledge of the structure of the system in question is required for its interpretation. In addition, while LZ allows for analysis of real systems based on time-series data, it is also the measure that is the furthest removed from IIT (but see [14]). It is highly dependent on the size of the input and is hard to interpret without normalization, which makes it difficult to compare systems of varying size. Finally, the measure Φ* is aimed at providing a tractable measure of integrated information using mismatched decoding and is applicable to time-series data, both discrete and continuous [10]. Φ* is relatively fast to compute and can also be applied to continuous time series like EEG. However, while we observed a high correlation with $\Phi_{3.0}^{peak}$, a cluster of high Φ* values with corresponding low $\Phi_{3.0}^{peak}$ values limited the interpretation. This suggests that Φ* might not be reliable for low $\Phi_{3.0}^{peak}$ networks, but the analysis of larger networks is needed to draw a conclusion. While the results did not suggest a clear tractable alternative to Φ_3.0_, several of the measures could be useful in statistical comparisons of groups of networks. 

Prior work directly comparing Φ_3.0_ with measures of differentiation (e.g., D1, LZ) reported lower correlations than those observed here for Φ_3.0_ [15]. There are at least three possible reasons for this: (a) the current work considered only linear nodes instead of nodes implementing general logic, (b) we compared against $\Phi_{3.0}^{peak}$ and not $\Phi_{3.0}^{mean}$, and (c) we considered only the whole system as a basis for the heuristics, and not the subset of elements that constitutes the MC. For (b), we reran the analysis replacing $\Phi_{3.0}^{peak}$ with $\Phi_{3.0}^{mean}$, producing negligible deviances in the results (see Section A.5). For (c), the results of the WS (whole-system approximation) suggested that using the whole system to approximate the MC does not make a substantial difference (at least for networks of this size). This leaves (a), the types of network studied, as the likely reason for the differences in the strength of the correlations. 

All heuristic measures’ rank correlations with $\Phi_{3.0}^{peak}$ were negatively impacted by removing networks with $\Phi_{3.0}^{peak}$ = 0‖1. This suggests that such networks are indeed relevant to consider and that finding a tractable measure that seperates $\Phi_{3.0}^{peak}$ = 0 and $\Phi_{3.0}^{peak}\geq \;0$ networks would be useful in its own right. Evident in the results was that all heuristics, except S, SI, and MI, showed an inverse predictability with $\Phi_{3.0}^{peak}$, i.e., low scores on a given heuristic corresponded to a low score on $\Phi_{3.0}^{peak},$ but the higher the scores, the larger the spread of $\Phi_{3.0}^{peak}$ (see Figure 5). This could explain why the correlations drop when removing networks with $\Phi_{3.0}^{peak}$ = 0‖1. This inverse predictability indicates two things. First, that the tested measures could be useful as negative markers, that is, low scores on measures can indicate low $\Phi_{3.0}^{peak}$ networks, but not the converse. Secondly, it suggests $\Phi_{3.0}^{peak}$ has dependencies on aspects of the underlying network that are not captured by any of the heuristic measures.

### 4.3. Future Outlook

Finally, we discuss several topics that we consider to be relevant for future work. First, there are several conceptual aspects of Φ_3.0_ that are worth considering when developing future methods. *Composition:* One of the major changes in IIT_3.0_ from previous iterations of the theory is the role of all possible mechanisms (subsets of nodes) in the integration of the system as a whole. To our knowledge, all existing heuristic measures of integrated information are wholistic, always looking at the system as a whole. Future heuristics could take a compositional approach, combining integration values from subsets of measurements, rather than using all measurements at once. *State dependence*: We report that heuristic measures do not correlate with state-dependent Φ_3.0_ (see Section A.6 for a perturbation-based approach), but a more accurate statement is that there are no (data-based) state-dependent heuristics; the nature of heuristic measures does not naturally accommodate state-dependence. *Cut directionality*: Φ_3.0_ uses unidirectional cuts, i.e., separating one directed connection, while other heuristics use bidirectional cuts (Φ_2.0_, Φ_2.5_) or even total cuts, separating system elements (Φ*, SI, MI). This leads, in effect, to an overestimation of integrated information, even for feedforward and ring-shaped networks (see Figure 2). This could potentially partially explain the inverse predictability noted above.

Secondly, there are differences in the data used for the different measures. Only the approximations (and D1/D2/Φ_2.0_/Φ_2.5_) were calculated on the full TPM, the other heuristics were calculated on the basis of the generated time-series data. However, while deterministic networks such as those considered here can be fully described by both time-series data and TPM, given that the system was initialized to all possible states at least once, data from deterministic systems might be “insufficient” as a time series, as they often converge on a few cyclical states and, as such, need to be regularly perturbed. One solution to this could be to add noise to the system to avoid fixed points. In addition, as all heuristics considered here (except D1/D2/Φ_2.0_/Φ_2.5_) were dependent on the size of the generated time series (see Section A.1), future work should control for the number of samples and discuss the impact of non-self-sustainable activity (convergence on a set of attractor states).

Thirdly, studies comparing measures of information integration, differentiation, and complexity, have also observed both qualitative and quantitative differences between the measures, even for simple systems [19,20]. Thus, there might be a large number of networks where the tested heuristics would correspond to Φ_3.0_ if only certain prerequisites are met, such as a certain degree of irreducibility or small-worldness. One could, for example, imagine systems that have evolved to become highly integrated through interacting with an environment [34]. Such evolved networks might have further qualities than being integrated, such as state differentiation that serves distinctive roles for the system, i.e., differences that make a behavioral difference to an organism, which is an important concept in IIT (although considered from an internal perspective in the theory) [5]. While it is still an open question what Φ_3.0_ captures of the underlying network above that of the heuristics considered here, investigation into structural and functional aspects that lead to systems with high Φ_3.0_ could point to avenues for developing new measures inspired by IIT. Further, while estimates of the upper bound of Φ_3.0_, given a system size, have been proposed (e.g., see [15]), not much is known about the actual distribution of Φ_3.0_ over different network types and topologies. Here, we explored a variety of network topologies, but the system properties, such as weight, noise, thresholds, element types, and so on, were omitted because of the limited scope of the paper. Investigating the relation between such network properties and Φ_3.0_ would be an interesting research project moving forward. This could be useful as a testbed for future IIT-inspired measures and be informative about what kind of properties could be important for high Φ_3.0_ in biological systems and the properties to aim for in artificial systems to produce “consciousness”. 

Finally, there are several approximations and heuristics not included in the present study [11,12,19,28,35,36,37,38,39,40], some of which are specifically applicable to time-series data [10,11,12,19,21,28,40]. Accordingly, the present work should not be considered an exhaustive exploration of Φ_3.0_ correlates. 

## Figures and Tables

**Figure 1 entropy-21-00525-f001:**
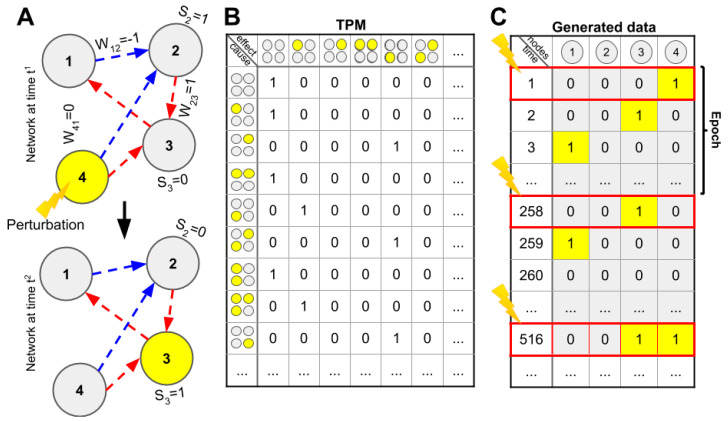
(**A**) Networks were randomly generated with *n* binary linear threshold nodes (S_i_ ∈ {0, 1}, ϴ ≥ 1.0) and connections (W_ij_ ∈ {−1, 0, 1}). Each network was perturbed into each possible initial state, and the following state transitions were recorded. (**B**) The networks’ node mechanism and connection weights were used to generate a transition probability matrix (TPM), containing the probability of one state leading to any other state. (**C**) From the TPM, we generated an “observed” time series using frequent perturbations of the initial states. The sequence of state transitions following an initial state perturbation is termed an epoch.

**Figure 2 entropy-21-00525-f002:**
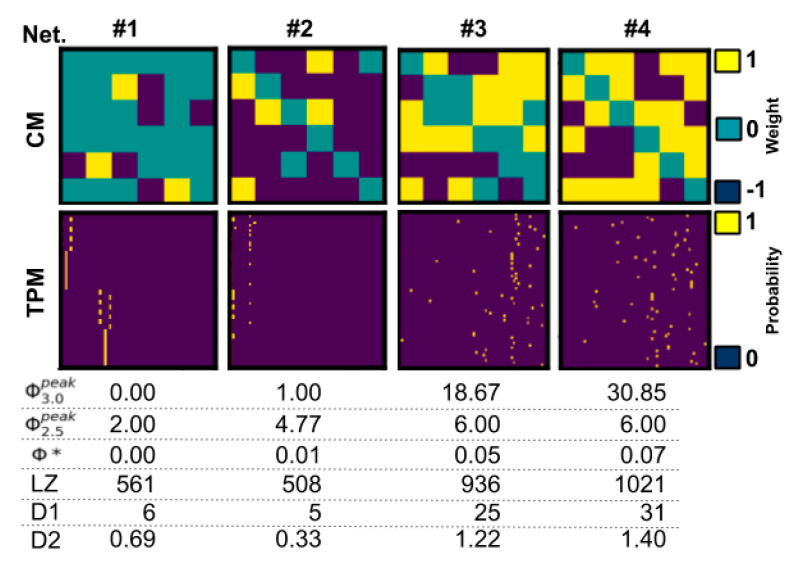
Four example networks with connection matrices (CM) and TPMs, with $\Phi_{3.0}^{peak}$ and corresponding values for selected state-independent heuristics. Note that network #1 does not consist of a feedforward network if you consider all connections in the CM but is a feedforward network if only excitatory (yellow) connections are considered, which is consistent with $\Phi_{3.0}^{peak}$ = 0. Network #2 consists of a simple ring-shaped network only if excitatory connections are considered, which is consistent with $\Phi_{3.0}^{peak}$ = 1.

**Figure 3 entropy-21-00525-f003:**
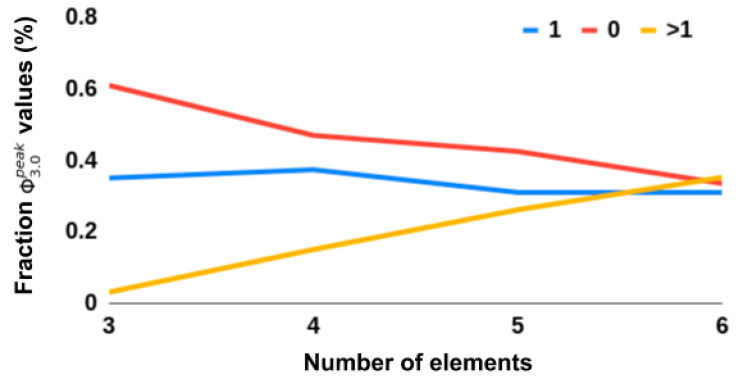
Overview of fraction of networks with $\Phi_{3.0}^{peak}$ ∈ {1, 0, >1}.

**Figure 4 entropy-21-00525-f004:**
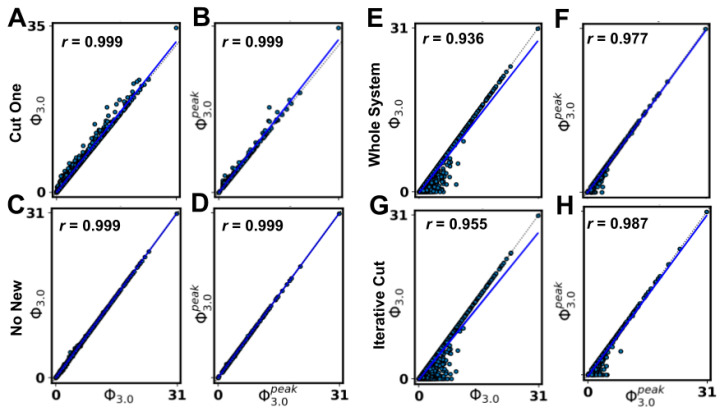
Results of the comparison between Φ_3.0_ and approximations, with plotted linear fit (blue) and one-to-one relationship (dotted, gray); (**A**) Φ_3.0_ of the state-dependent CO approximation, (**B**) $\Phi_{3.0}^{peak}$ of the state-independent CO, (**C**) Φ_3.0_ of the state-dependent NN approximation, (**D**) $\Phi_{3.0}^{peak}$ of the state-independent NN. (**E**) Φ_3.0_ of the state-dependent WS estimated main complex, (**F**) $\Phi_{3.0}^{peak}$ of the state-independent WS, (**G**) Φ_3.0_ of the state-dependent IC estimated main complex, (**H**) $\Phi_{3.0}^{peak}$ of the state-independent IC.

**Figure 5 entropy-21-00525-f005:**
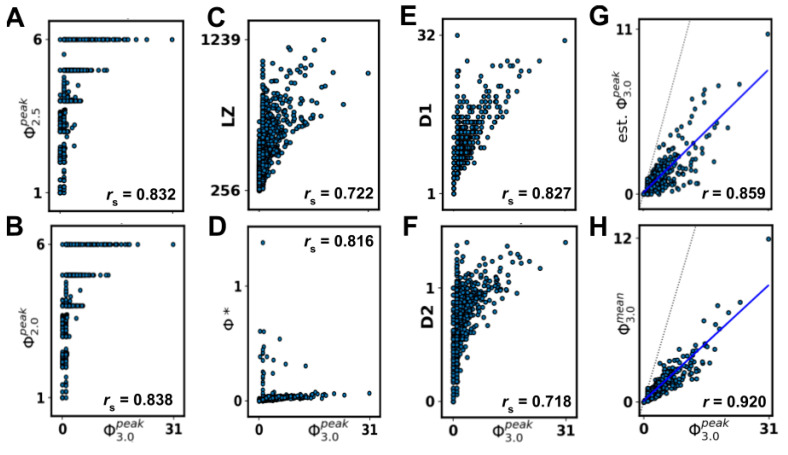
Results of comparison between state-independent $\Phi_{3.0}^{peak}$ and heuristics and estimates of $\Phi_{3.0}^{peak}$. (**A**) Φ_2.5_ modified from Φ_2.0_, (**B**) Φ_2.0_ based on IIT_2.0_, (**C**) LZ complexity (non-normalized), (**D**) decoder-based Φ, based on Φ_2.0_, (**E**) state differentiation D1, (**F**) cumulative variance of system elements D, (**G**) estimated state-independent $\Phi_{3.0}^{peak}$ using five randomly sampled states (**H**) state-independent $\Phi_{3.0}^{mean}$. **G** and **H** are plotted with linear fit (blue) and one-to-one relationship (dotted, gray).

**Table 1 entropy-21-00525-t001:** Overview of measures.

#	S.D. Measure	S.I. Measure	Description	Ref.
	Φ_3.0_	$\Phi_{3.0}^{peak}$	Integrated information according to IIT 3.0	[5]
A	CO Φ_3.0_	CO $\Phi_{3.0}^{peak}$	Cut one connection when making partitions	[6]
B	NN Φ_3.0_	NN$\Phi_{3.0}^{peak}$	No new concepts after partitioning	[6]
C	WS Φ_3.0_	WS$\Phi_{3.0}^{peak}$	Whole system as MC	
D	IC Φ_3.0_	IC$\Phi_{3.0}^{peak}$	Elements with recurrent connections as MC	
E		Est.*n*$\Phi_{3.0}^{peak}$	Estimate $\Phi_{3.0}^{peak}$ from n states (*n*=1,2,...,15)	
F	Φ_2.0_	$\Phi_{2.0}^{peak}$	Integrated information according to IIT 2.0	[3]
G	Φ_2.5_	$\Phi_{2.5}^{peak}$	Φ_2.0_/Φ_3.0_ hybrid	[12]
H		D1	Reachable states	[15]
I		D2	Cumulative variance of elements	[15]
J		*S*	Coalition sample entropy	[13]
K		LZ	Functional complexity	[13]
L		Φ*	Decoder based integrated information	[10]
M		SI	Integrated stochastic interaction	[11]
N		MI	Mutual information	[21]

**Abbreviations:** S.D.: state-dependent; S.I.: state-independent; Ref: reference; IIT: integrated information theory; Φ: integrated information; Φ^peak^: maximum Φ over system states; CO: cut-one approximation; NN: no-new-concepts approximation; WS: whole-system approximation; MC: major complex; IC: iterative-cut approximation; Est.*n*: $\Phi_{3.0}^{peak}$ estimated from *n* sample states; D1/2: state differentiation; *S*: coalition entropy; LZ: Lempel–Ziv complexity; Φ*: decoder-based Φ; SI: stochastic interaction; MI: mutual information.

**Table 2 entropy-21-00525-t002:** Overview of results.

#	S.D. Measure	*r*	S.I. Measure	*r*
	Φ_3.0_		$\Phi_{3.0}^{peak}$	
A	CO Φ_3.0_	0.999	CO $\Phi_{3.0}^{peak}$	0.999
B	NN Φ_3.0_	0.999	NN$\Phi_{3.0}^{peak}$	0.999
C	WS Φ_3.0_	0.936	WS$\Phi_{3.0}^{peak}$	0.977
D	IC Φ_3.0_	0.955	IC$\Phi_{3.0}^{peak}$	0.987
E			Est_5_Φ_3.0_	0.859
F	Φ_2.0_	0.622	$\Phi_{2.0}^{peak}$	0.838
G	Φ_2.5_	0.473	$\Phi_{2.5}^{peak}$	0.832
H			D1	0.827
I			D2	0.718
J			*S*	0.711
K			LZ	0.722
L			Φ*	0.816
M			SI	0.537
N			MI	0.306

**Abbreviations:***r*: correlation values, with measures A–F using Pearson’s *r*, and G–O using Spearman’s *r*_s_; S.D.: state-dependent; S.I.: state-independent; Φ: integrated information; Φ^peak^: maximum Φ over system states; CO: cut-one approximation; NN: no-new-concepts approximation; WS; whole-system approximation; IC: iterative-cut approximation; Est_5_: $\Phi_{3.0}^{peak}$ estimated from five sample states; D1/2: state differentiation; *S*: coalition entropy; LZ: Lempel–Ziv complexity; Φ*: decoder-based Φ; SI: stochastic interaction; MI: mutual information.

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
