# Peer review of "Evaluating Approximations and Heuristic Measures of Integrated Information"

_entropy, 2019, doi:10.3390/e21050525_

Reviewer 1 Report

Summary

The measure proposed with IIT, Phi3.0, has extremely high computational costs (among other issues with applicability to data). Consequently, approximations and heuristics have been put forward. Here, the authors investigate the convergent validity of some of the popular approximations and heuristics with Phi3.0 and report generally high validity of approximations which do not reduce computation times much but lower validity of cheaper heuristics. These results will likely be useful for future empirical research in choosing the best heuristics for integrated information to use. I like this paper in general and I recommend its acceptance in principle. However, various parts of the manuscripts are unclear (or potentially include some mistakes). Please see my detailed comments below.

Signed review by Nao Tsuchiya (with discussion with my PhD candidate student, Angus Leung)

Major issues

Meaning of state dependent values for phi*, SI, MI

Phi*, SI, and MI are inherently state-independent, so it’s unclear what you get out of your particular way of defining state dependent values for Phi*, SI, and MI.

One way to interpret these is to consider them in the context of empirical applications, like TMS-EEG experiments. By applying these measures to the EEG response after one TMS pulse (as perturbation), you can presumably compute these values. Then we can interpret these as a potential index of level of consciousness.  

If you interpret them in this way, however, I find your way of computing these values to be problematic. In Haun et al 2017 eNeuro, for example, we assessed how patterns of evoked phi* are correlated with contents of consciousness. There, we computed phi* (across many trials) but for a short period of the time (200 ms). This short period of the time makes sense, also for TMS-EEG experiments. A TMS-EEG paper (Casali et al 2013) uses only initial part of the EEG response to TMS for their PCI measure.  If you are using 2^6 - 2^9 data points  (As I mention later, I don’t understand exactly how many time points you used for each state dependent measure), I think what you are computing as state dependent values are at the regime where perturbations are mostly irrelevant.

Can you comment on this?  

Further, there are some technical issues with phi*. First, which partition scheme did you use?  Did you use the atomic partition?  Did you search for the MIP? If so, how did you compare partitions (presumably using normalisation as specified in Balduzzi and Tononi, 2008 PLOS Computational Biology)? I think Practical Phi toolbox v1.0 computes phi-star using the Gaussian approximation.  This needs to be explicitly stated.  Treating binarized values as Gaussian variables may have weakened the correlation with Phi3.0.

2. Perturbation

Page 3, para 2: Can you give some more explanation as to: 1) how you perturb the system to construct STM, 2) how you translate STM into TPM, and 3) how you use STM to compute state independent values of K-O?

Initially, I was confused why you perturb the system first with [0100], then [0001], then [0110] (in Figure 1B) because you eventually perturb the system in all possible ways. Why did you randomize the order of perturbation?  I guess you did this because 1) when you translated STM into TPM, you didn’t count the transition from the end of cycle into perturbed state (say, from 4th to 5th row in Fig 1B).  However, 2) when you computed state independent values of K-O, you seemed to have used all temporal transition, from 1 to 2, 2 to 3, …. including 4-1 transition. Is this correct?  

Can you be more explicit about this?  

3. Adjustment factor

I don’t understand the global logic and detailed procedures and formulas for the adjustment factor.

First as to the global logic.  In p3, you say, “adjustment factor … avoids normalization issues between systems of different number of nodes".  

I can imagine two arguments for normalization. As in Figure 1B, if you have 4 timesteps for 4 neurons, then one normalization scheme would be to use the same number of data points for a 5 neuron case.  (I guess this is similar to what you mean by “force the same number of bits” - which I don’t fully understand.) In this case you would propose something like 4 x 4 / 5 = 3.2 rows for 5 neuron case.  

Another possible normalization scheme would be that , if you have more neurons, you supply more data (which seems also a natural assumption).

As I describe below, due to the unclear nature of the equations associated with this factor, I’m not sure which one you intended.

Can you clarify your logic on why you chose one over the other (or any other options)?  

Second as to the detailed procedures and equations.

Assuming that you are suggesting the former, divisive normalization scheme, I don’t understand equation 2. Can you give the logic?

The factor you give in Appendix is 2(n)n2^(6n) while that you give in main text is alpha(n)n2^(6+2). Which one is correct?

As to 2), I assume that Appendix version is correct. If I substitute nmax = 6, then (2) becomes smax = 6*2^12. By substituting this into (1), I get alpha(6) = 6*2^12 / 6^2 * 2^(6*7)

Then, alpha(6)*6*2^(6*6)=(2^48) / 2(^42) = 2^6.   For 3, 4, and 5, neurons, we get 2^9, 2^8, and 2^7 samples. This is consistent with the first philosophy of normalization.  

However, this Appendix version does not make sense, according to the logic of the main text in page 3. There, you say, “(sequence of states) resulted in alpha(n) * n * 2^(6+n)”.  n factor is the # of nodes = columns. (I suggest you rewrite this into “STM with n columns x alpha(n) *2^(6+n) rows”)

Now, 2^n factor comes from all possible ways to perturb the system.  So, the row can be further divided into:  

2^n perturbations x alpha(n)*2^6.  

In this scheme, I can understand that 2^6 factor is the minimum amount of the data you want to have for 6 neuron case (correct?).

But this logic then leads to the alpha equation:

n * 2^n * 2^6 * alpha(n) = n* 2^(n+6) * alpha(n).

If I use this, then I get (2^24) / (2^42) < 1 time steps for 6 neurons.  There is something wrong.

Relatedly, I suggest that in Appendix A1 equation 2, you include the actual value of smax (=6) to make it clear what you really ended up with.  Also, maybe worth adding a summary of the effect of alpha(n) (e.g. for larger n, fewer timesteps are simulated than just 2^6).

Please clarify this issue.

4. Meaning of phi3.0max

From IIT, I can imagine three possible state independent measures of level of consciousness: 1) Phi3.0max, 2) Phi3.0mean (across states without any weighting, which you report in Appendix and Fig 5D. Can you mention this more explicitly early in the manuscript?) and 3) Phi3.0mean (weighted across actual states, as used in Albantakis et al 2014).

You need to give theoretical justification for why you chose 1 over the others. 3) is not theoretically coherent with IIT as it requires an extrinsic observer who tallies actual state history.  1) and 2) are possible, but you need to discuss problems in each case. 1) is strange to most people (I imagine). Imagine a system, in which it gives very low big phi values for most of the possible states. But this system gives one very high big phi values in only one state. 1) will say that this system is highly conscious. For the same system, 2), on the other hand, will say that the system is nearly not conscious.

Please comment on this issue.

(Empirically, you show high correlation in Fig 5D, so it’s not a big issue, but the interpretation of Phi3.0max can be very problematic or counter-intuitive to many people).  

5. Iterative cutting & meaning of on and off

Regarding iterative cut (approximation E), this requires justification and needs some discussion on problems associated with this.  In principle, the labeling of on vs off is arbitrary and there is no way to tell the difference between them from an intrinsic viewpoint.  In this paper, because you externally introduced a threshold of +1, intrinsically, you can claim that you can remove always-off nodes as not “causally reachable”.

However, if you were to introduce a concept of “causal reachability”, then shouldn’t you also introduce it when you apply “perturbation” per se? In fact, a pattern of neural activity evoked by a single TMS pulse is mostly causally unreachable in this sense, but you are using it to construct both STM and TPM. So, theoretically, I think it’s incoherent.   

This is potentially quite philosophical and/or technical, but I think this deserves some discussion (possibly in footnote).

6. Statistics

As you declared in the methods, given a huge number of networks tested, I don’t think you need to report p values (also F values), not only for Spearman, but also for Pearson.  

Assuming you do that in the revision, I’m a bit confused by the way the results are reported in Section 3.2. You say “Regression analysis showed … stronger …” but I don’t see any direct statistical comparisons. If you remove unnecessary p- and F- values, it’ll become clear that you never did proper comparisons, correct?  

You can use linear mixed effects models (or any other methods) to do such a direct comparison.  (Or you can remove the comparative statement).  

Minor issues

Page 4.  “earth’s mover distance” -> “earth mover’s distance”

Table A1.  Can you replace r and rs with delta_r and delta_rs to make it clear that they are the difference from the original values?

Page 5.  “tor” -> “for”

Page 8 para 3: “through Lempel Ziv compression (LZ), indicating a the degree of order or patterns…” –> remove ‘a’ or ‘the’

Page 8 para 5: Suggest that you change  

“As the STM consisted of series of simulated state sequences starting from a perturbation into every possible state of the system, 2^n, …” into

“As the STM consisted of series of simulated state sequences starting from each of 2^n possible perturbed states of the system, …”

page 8 n(1981) -- inconsistent with 2032 reported in page 9. (Actually the meaning of n(1981) is not very clear in page 8). Given the F stats that are reported later, I guess n was actually 1981.

Page 9 para 2: “PyPHI” -> “PyPhi”

Page 16 para 1: “butthe” -> “but the”

Page 16 para 3: Unclear what you meant in the following sentence

“This suggests that such networks are indeed relevant to consider as **both that** finding a tractable measure that…”

Page 17 para 2: Clarify - how does IIT use the “differences that make a difference to an organism” differently? Is the emphasis here to do with differences which have a functional purpose for the organism? It’s not clear from the way it’s currently written. (Also, grammar, “differences that make”, not “differences that makes”)

Page 17 para 2: One outstanding difference worth mentioning is composition, which to my knowledge is implemented only in phi-3

Page 17 para 3:

“testbed for future IIT inspired measure…” ->

“testbed for future IIT inspired measures…” or

“testbed for a future IIT inspired measure”

Appendix A1 para 1:

“Each network were perturbed” -> “Each network was perturbed”

“is calculate” -> “are calculated”

Appendix A4 para 1: “The average times was recorder…” -> “The average times were recorded…”

Appendix A4: Could you report the computational resources used (and if possible, summary of degree of parallelisation implemented / effective resources used for each measure; but likely it is the case that only pyphi implements parallelisation)?

Table A2: Formatting error at end of caption? “all: measure applied to the whole state transition matrix (STM)”

Figure A3: Can add white outlines to each triangle to help visually differentiating close points?

Author Response

Response to reviewer #1:

We thank the reviewer for the their thorough review which helped us a lot in improving the manuscript. In response to general issues raised, we have performed the following broad changes which should increase readability, conciseness, as well as remove points of confusion. (1) state transition matrices (STM) has been reworded into “observed” data (with perturbation), along with a rewrite of section 2.1 concerning TPMs and time series data, and how they are generated. (2) state dependent heuristics on observed data has been reformulated as “initial state dependent heuristics” and moved to appendix to avoid confusion with the standard notion of state dependence in IIT. (3) The All Partitions “approximation” has been moved to an appendix and mentioned in the discussion. We acknowledge that it’s not an approximation, but rather a potential revised definition for the MIP.

Following is a point by point treatment of the specific comments raised. Changes in the manuscript are marked by yellow.

1. State dependent heuristics

Meaning of state dependent values for phi*, SI, MI

Phi*, SI, and MI are inherently state-independent, so it’s unclear what you get out of your particular way of defining state dependent values for Phi*, SI, and MI.

One way to interpret these is to consider them in the context of empirical applications, like TMS-EEG experiments. By applying these measures to the EEG response after one TMS pulse (as perturbation), you can presumably compute these values. Then we can interpret these as a potential index of level of consciousness.  

If you interpret them in this way, however, I find your way of computing these values to be problematic. In Haun et al 2017 eNeuro, for example, we assessed how patterns of evoked phi* are correlated with contents of consciousness. There, we computed phi* (across many trials) but for a short period of the time (200 ms). This short period of the time makes sense, also for TMS-EEG experiments. A TMS-EEG paper (Casali et al 2013) uses only initial part of the EEG response to TMS for their PCI measure.  If you are using 2^6 - 2^9 data points  (As I mention later, I don’t understand exactly how many time points you used for each state dependent measure), I think what you are computing as state dependent values are at the regime where perturbations are mostly irrelevant.

Can you comment on this?  

First, we clarify what is meant by “state dependent” for these heuristics. These measures are state-dependent in the sense that they are computed on observed data starting from a particular initial state (without perturbation into other states). A more accurate description would be “initial-state-dependent”. We have decided to move these initial state dependent heuristics (including phi*, SI, MI) to Appendix A6 to avoid confusion, as this notion is very different from the notion of state dependence for Φ3.0 and the various approximations. Accordingly, changes have been made throughout the manuscript, in addition to a new Appendix section (A6).

As you suggest, these measures are based on an initial condition/perturbation similar to (Casali et al., 2013) or (Haun et al., 2017). One way to use this data is to test whether observed data from an initial condition, can be used to estimate state dependent Φ3.0 (our results suggested that in general they cannot, see Appendix A6). Another use for these data is to look at the variability in Φ3.0 estimates, to determine the benefits of a perturbational approach vs. a purely observational approach. In either case, how many timesteps are needed to be analysed/generated after a perturbation/initial condition is an open question which we don’t have an answer to. For example, one timestep for a discrete binary system is ambiguous with respect to what that means for e.g. EEG data (and vice versa). So how many timesteps/samples are “sufficient”? What is the time equivalent of a single sample in an abstract logic gate network?

Since we mainly aim to test state independent heuristics based on observed data, we needed to have the same amount of data in all matrices across different network size (hence the function ɑ(n), Appendix A1). This choice was based on the fact that Lempel Ziv compression is heavily dependent on the size of its input, although this was also the case for the other heuristics (Appendix A1, Figure A1). As such, networks with fewer nodes need to be simulated for longer to ensure parity of bits in the entire network. Given that the minimum amount of states needed to be generated following an initial state is 2n (ensure possibility of visiting the full state space), this causes, in almost all cases, that each epoch is too long and thus enter a set of attractor states.

Although, the takeaway from the issues raised, and results presented in Appendix A1, is that with these measures, care should be taken when it comes to number of datapoints included in the analyses. A line to this effect has been added to the discussion.

Added to section 4.3, paragraph 2:

“While in deterministic networks, the time series data and TPM contain the same information, given that the system is initialized to all possible states at least once, the data generated from the types of networks considered here might be “insufficient” as a time series, e.g. systems often converge on a few cyclical states and as such needs to be regularly perturbed to avoid long stretches of uninformative sequences of states. As all heuristics considered here were dependent on the size of the generated time series (see Appendix A1), future work should control for the number of samples and discuss the impact of non-self-sustainable activity (convergence on a set of attractor states).”

2. Technical issues with Phi*

Further, there are some technical issues with phi*. First, which partition scheme did you use?  Did you use the atomic partition?  Did you search for the MIP? If so, how did you compare partitions (presumably using normalisation as specified in Balduzzi and Tononi, 2008 PLOS Computational Biology)? I think Practical Phi toolbox v1.0 computes phi-star using the Gaussian approximation.  This needs to be explicitly stated.  Treating binarized values as Gaussian variables may have weakened the correlation with Phi3.0.

Technical parameters that were missing in the methods section have now been added in section 2.3.3, paragraph 2. Specifically, we employed a bipartition scheme over the full powerset 2n-1-1, searched for the MIP using an exhaustive search, employed the variants defined for discrete data, and used the normalization scheme presented in Balduzzi and Tononi (2008).

Added to section 2.3.3:

“... with the discrete forms of the formulae, employing a MIP exhaustive search with a bipartition scheme (powerset; 2n-1-1), and normalization factor according to IIT2.0 [3]. “

3. Perturbation, STM, and TPM

Page 3, para 2: Can you give some more explanation as to: 1) how you perturb the system to construct STM, 2) how you translate STM into TPM, and 3) how you use STM to compute state independent values of K-O?

Initially, I was confused why you perturb the system first with [0100], then [0001], then [0110] (in Figure 1B) because you eventually perturb the system in all possible ways. Why did you randomize the order of perturbation?  I guess you did this because 1) when you translated STM into TPM, you didn’t count the transition from the end of cycle into perturbed state (say, from 4th to 5th row in Fig 1B).  However, 2) when you computed state independent values of K-O, you seemed to have used all temporal transition, from 1 to 2, 2 to 3, …. including 4-1 transition. Is this correct?  

Can you be more explicit about this?

Generally, as the formulation of STM was confusing, this has now been renamed to observed time series data. As a response to the raised issues, section 2.1 has been partially rewritten to reflect better how the TPM and “observed” (STM) data were generated, an edit to section 2.3.3, as well as edits to Figure 1, and a complete revision of Appendix A1.

Specifically, (1) perturbations happened by programmatically setting the current state every ɑ(n)(2n + 1) timesteps (see Appendix A1, and our response to point 4. below), sequencing through all possible states of the system. (2) this was unspecific, and for clarity the TPM was generated based on the system architecture (however, as these are deterministic systems, observed data and the TPM contain the same information). (3) we used the entire observed data (STM) as if it was a long time series with regular perturbations.

Regarding order of perturbations, we perturbed the system into all possible states, using the “LOLI” format used in the PyPhi software (see Mayner et al (2018)).

Regarding counting transitions from end of cycle, transitions into perturbations were not excluded for the state independent heuristics that used the entire observed time series. All other measures used the TPM which is independent of the observed time series, and thus did not take into account transitions into perturbed states.

Partial rewrite of section 2.1:

“To investigate various measures and approximations we need functional information about the networks in the form of a probabilistic description of the transitions from any given state to any other state, i.e., a transition probability matrix (TPM). For each network, the TPM was constructed based on the node mechanism (linear threshold with ϴ = 1) and the connection weights Wij. As the generated networks were deterministic, the TPM contained only a single ‘1’ in each row representing the next state of the network.

From the TPM, given an initial condition, we were able to generate “observed” time series data for each network. From a given initial condition, a network my only explore part of its state space before reaching an attracting fixed point or periodic sequence. While generating the observed data, we periodically perturbed the network into a new state, ensuring that our data fully explored the state space of the network, and that the results were not dependent on our choice of initial condition. This procedure resembles the perturbation applied by TMS-EEG during empirical studies of consciousness (Casali et al., 2013). The generated time series data consisted of 2n epochs, where one epoch was generated by initializing/perturbing a network to an initial state, then simulated for a total of ɑ(n)(2n + 1) timesteps. The function ɑ(n) ensures parity of bits between the generated time series for networks of different sizes (see Appendix A1). This perturbation & simulation process was repeated for all possible network states (2n) sequentially, with each epoch appended to the last preceding epoch. The resulting simulated time series (sequence of epochs) produced a ɑ(n)(2n + 1)2n-by-n matrix where each of n columns reflects the state of a single node over time, and each row reflects the current state of each network node (0/1) at a given time. In sum, we derive a TPM from the mechanism and connectivity profile of individual nodes, and then using the TPM and perturbations, generate a time series of observed data that explores the entire state space of the network. See Figure 1b,c.“

Added to section 2.3.3

“All heuristics were calculated in a state-independent manner, using the time series data generated for the whole network (no searching through subsystems).”

4. Adjustment factor

I don’t understand the global logic and detailed procedures and formulas for the adjustment factor.

First as to the global logic.  In p3, you say, “adjustment factor … avoids normalization issues between systems of different number of nodes".  

I can imagine two arguments for normalization. As in Figure 1B, if you have 4 timesteps for 4 neurons, then one normalization scheme would be to use the same number of data points for a 5 neuron case.  (I guess this is similar to what you mean by “force the same number of bits” - which I don’t fully understand.) In this case you would propose something like 4 x 4 / 5 = 3.2 rows for 5 neuron case. Another possible normalization scheme would be that , if you have more neurons, you supply more data (which seems also a natural assumption). As I describe below, due to the unclear nature of the equations associated with this factor, I’m not sure which one you intended. Can you clarify your logic on why you chose one over the other (or any other options)?

Second as to the detailed procedures and equations.

Assuming that you are suggesting the former, divisive normalization scheme, I don’t understand equation 2. Can you give the logic? The factor you give in Appendix is 2(n)n2^(6n) while that you give in main text is alpha(n)n2^(6+2). Which one is correct? As to 2), I assume that Appendix version is correct. If I substitute nmax = 6, then (2) becomes smax = 6*2^12. By substituting this into (1), I get alpha(6) = 6*2^12 / 6^2 * 2^(6*7). Then, alpha(6)*6*2^(6*6)=(2^48) / 2(^42) = 2^6.   For 3, 4, and 5, neurons, we get 2^9, 2^8, and 2^7 samples. This is consistent with the first philosophy of normalization.  However, this Appendix version does not make sense, according to the logic of the main text in page 3. There, you say, “(sequence of states) resulted in alpha(n) * n * 2^(6+n)”.  n factor is the # of nodes = columns. (I suggest you rewrite this into “STM with n columns x alpha(n) *2^(6+n) rows”). Now, 2^n factor comes from all possible ways to perturb the system.  So, the row can be further divided into:  2^n perturbations x alpha(n)*2^6.  In this scheme, I can understand that 2^6 factor is the minimum amount of the data you want to have for 6 neuron case (correct?). But this logic then leads to the alpha equation: n * 2^n * 2^6 * alpha(n) = n* 2^(n+6) * alpha(n).

If I use this, then I get (2^24) / (2^42) < 1 time steps for 6 neurons.  There is something wrong.

Relatedly, I suggest that in Appendix A1 equation 2, you include the actual value of smax (=6) to make it clear what you really ended up with. Also, maybe worth adding a summary of the effect of alpha(n) (e.g. for larger n, fewer timesteps are simulated than just 2^6).

Please clarify this issue.

There were many issues concerning the adjustment factor which was primarily caused by the equations being wrong. We thank the reviewer for raising the issue and allowing us to correct it. This has now been updated in both Appendix A1, and in section 2.1. See also our response to 1 and 3. We note that the purpose of the adjustment factor is to ensure that the size of the observed data is the same across network sizes. We hope that the following corrections are sufficient to illuminate the size of the observed data set.

Section 2.1:

“The generated time series data consisted of 2n epochs, where one epoch was generated by initializing/perturbing a network to an initial state, then simulated for a total of ɑ(n)(2n + 1) timesteps. The function ɑ(n) ensures parity of bits between the generated time series for networks of different sizes (see Appendix A1). This perturbation & simulation process was repeated for all possible network states (2n) sequentially, with each epoch appended to the last preceding epoch. The resulting simulated time series (sequence of epochs) produced a ɑ(n)(2n + 1)2n-by-n matrix where each of n columns reflects the state of a single node over time, and each row reflects the current state of each network node (0/1) at a given time.”

Appendix A1:

A1. Input size

For each network, N with n∈{3,4,5,6} elements, we generated an observed time series as a matrix AN, consisting of n columns, and m rows. To cover the full state space of N, we needed to perturb each N into each of the 2n possible initial conditions Si. For each initial condition Si we wanted to cycle through at least 2n+1 states (referred to as an epoch) to ensure that we simulate the full behavior of the network on each initial state Si, even for the longest possible cycle of length 2n. Thus, AN is a matrix of at least size n x m(n), where,

m(n) = (2n + 1)2n

However, as the LZ compression is dependent on the amount of data to compress, we wanted the size of AN to be equal for all n. Hence we needed to adjust the number of timesteps that we ran the simulation for so that the size of AN would always be the same as the largest network in the set, ň. Thus, for the specific case of ň = 6, the size of AN is given by,

ň x m(ň) = 6 x m(6) = 24960

To get the same size of AN for a network N of size n∈{3,4,5,6}, we need an adjusted number of timesteps m’(n) ≈ ɑ(n) x m(n) (rounded to the nearest integer) where the adjustment factor ɑ(n) is given by:

n x ɑ(n) x m(n) = 24960

                                   ɑ(n) = 24960 / n(2n+1)2n

For the general case, the shape of AN is n - by - m’(n) where  

m’(n) ≈ ɑ(n) x m(n)

m(n) = (2n + 1)2n

ɑ(n) = ň(2ň + 1)2ň / n(2n+1)2n

where n ∈ {a, a+1,..., ň}, for some a,ň ∈ ℕ, with ň > a.

5. Meaning of Φ3.0max

From IIT, I can imagine three possible state independent measures of level of consciousness: 1) Phi3.0max, 2) Phi3.0mean (across states without any weighting, which you report in Appendix and Fig 5D. Can you mention this more explicitly early in the manuscript?) and 3) Phi3.0mean (weighted across actual states, as used in Albantakis et al 2014).

You need to give theoretical justification for why you chose 1 over the others. 3) is not theoretically coherent with IIT as it requires an extrinsic observer who tallies actual state history.  1) and 2) are possible, but you need to discuss problems in each case. 1) is strange to most people (I imagine). Imagine a system, in which it gives very low big phi values for most of the possible states. But this system gives one very high big phi values in only one state. 1) will say that this system is highly conscious. For the same system, 2), on the other hand, will say that the system is nearly not conscious.

Please comment on this issue.

(Empirically, you show high correlation in Fig 5D, so it’s not a big issue, but the interpretation of Phi3.0max can be very problematic or counter-intuitive to many people).  

We interpret the maximum value across states (3.0peak) as a measure of a network’s capacity for consciousness. We expect that this value is strongly related to the average Φ3.0 value (equally weighted over all states), but less related to the time-averaged value. To clarify this issue, we’ve added the following paragraph to section 2.2:

As many of the heuristics and approximations outlined below are state-independent, there is no direct comparison to the state-dependent Φ3.0. To facilitate comparisons with these measures, we further compute a state-independent quantity, 3.0peak, as the maximum value of Φ3.0 across all states of the network. The quantity 3.0peakcan be thought of as a measure of a networks capacity for consciousness, rather than its currently realized level of consciousness. Alternatively, we could also compute the mean value of Φ3.0, which has some relation to the state-dependent value of Φ3.0 under certain regularity conditions [15], but the results were similar (see Figure 5d).

6. Iterative cutting approximation

Regarding iterative cut (approximation E), this requires justification and needs some discussion on problems associated with this.  In principle, the labeling of on vs off is arbitrary and there is no way to tell the difference between them from an intrinsic viewpoint.  In this paper, because you externally introduced a threshold of +1, intrinsically, you can claim that you can remove always-off nodes as not “causally reachable”.

However, if you were to introduce a concept of “causal reachability”, then shouldn’t you also introduce it when you apply “perturbation” per se? In fact, a pattern of neural activity evoked by a single TMS pulse is mostly causally unreachable in this sense, but you are using it to construct both STM and TPM. So, theoretically, I think it’s incoherent.   

This is potentially quite philosophical and/or technical, but I think this deserves some discussion (possibly in footnote).

We’ve rewritten the relevant paragraph in section 2.3.1 to better explain this, as well as a footnote. First, we do not mean to attach any specific meaning to the states “ON” and “OFF” (as these are indeed arbitrary). It is just the case that the condition we describe only happens for the “OFF” state in our networks. Moreover, the assumption is that we can always perturb a node into any state, and the IC approximation is only identifying nodes that have no “intrinsic” way to change state (i.e., there is no state of the system that will cause them to change).

Added to section 2.3.1:

We also tested two approximations based on a priori assumptions about which nodes are included in the MC. These approximations assumed the MC contained C) all the nodes in the system taken as a whole (whole system; WS), or D) the subsystem of the network where all nodes with no recursive connectivity (no input and/or output connections) or an unreachable state (nodes that are always “on” or always “off”1, such as a node with only inhibitory inputs) have been removed, iteratively (iterative cut; IC). In IIT3.0 such a node (either with no inputs, no outputs, or an unreachable state) can be partitioned without loss, leading to Φ3.0 = 0. Simply excluding these nodes from the MC is not an approximation but a computational short-cut, as they will necessarily be outside the MC. However, assuming that the remaining set of recursively connected nodes is the MC is the approximation.

Footnote:

1 By unreachable, we mean there is no “internal” state of the network that will lead to the particular node being “on” or “off” in the next time step. This is does not mean that we cannot use an external perturbation to set the node into any state (which we do when generating the observed data).

7. Statistics

As you declared in the methods, given a huge number of networks tested, I don’t think you need to report p values (also F values), not only for Spearman, but also for Pearson.  

Assuming you do that in the revision, I’m a bit confused by the way the results are reported in Section 3.2. You say “Regression analysis showed … stronger …” but I don’t see any direct statistical comparisons. If you remove unnecessary p- and F- values, it’ll become clear that you never did proper comparisons, correct?  

You can use linear mixed effects models (or any other methods) to do such a direct comparison.  (Or you can remove the comparative statement).

As suggested, we have now removed unnecessary p- and F- values throughout results (section 3). In addition, we removed mention of direct statistical comparisons between heuristics and approximations, i.e. “... stronger… “, in section 3.2, as this was based on observations rather than specific tests, and are not relevant for the research question.

8. Minor issues

We thank the reviewer for finding these mistakes and errors. We have now corrected them and made necessary adjustments. We will only go through them in brief and only add commentary where necessary.

Page 4.  “earth’s mover distance” -> “earth mover’s distance” - Fixed

Table A1.  Can you replace r and rs with delta_r and delta_rs to make it clear that they are the difference from the original values? - Fixed

Page 5.  “tor” -> “for” - Fixed

Page 8 para 3: “through Lempel Ziv compression (LZ), indicating a the degree of order or patterns…” –> remove ‘a’ or ‘the’ - Fixed

Page 8 para 5: Suggest that you change  “As the STM consisted of series of simulated state sequences starting from a perturbation into every possible state of the system, 2^n, …” into “As the STM consisted of series of simulated state sequences starting from each of 2^n possible perturbed states of the system, …”

Irrelevant after rewrite.

page 8 n(1981) -- inconsistent with 2032 reported in page 9. (Actually the meaning of n(1981) is not very clear in page 8). Given the F stats that are reported later, I guess n was actually 1981.

Fixed. 2032 is the total number of networks, however, 51 of the six node networks were not analyzed with the full range of measures due to a computer crash and corrupted data. Regeneration of these was too costly in terms of time given submission deadline. However, this can be done if so desired, but it has negligible impact on results.

Page 9 para 2: “PyPHI” -> “PyPhi” - Fixed

Page 16 para 1: “butthe” -> “but the” - Fixed

Page 16 para 3: Unclear what you meant in the following sentence “This suggests that such networks are indeed relevant to consider as **both that** finding a tractable measure that…”

Fixed. This now reads:This suggests that such networks are indeed relevant to consider and that finding a tractable measure that separates 3.0peak= 0 and 3.0peak= > 0 networks would be useful in its own right.”

Page 17 para 2: Clarify - how does IIT use the “differences that make a difference to an organism” differently? Is the emphasis here to do with differences which have a functional purpose for the organism? It’s not clear from the way it’s currently written. (Also, grammar, “differences that make”, not “differences that makes”)

Here we implied that certain states would have behavioral differences that made a difference to the system, i.e. meaningful behavior in terms of fitness. In IIT terms, this is considered from an internal perspective, which is needed for behavioral differentiation anyway. The corresponding sentence has been altered to:differences that make a behavioral difference to an organism, which is an important concept in IIT (although considered from an internal perspective in the theory)”

Page 17 para 2: One outstanding difference worth mentioning is composition, which to my knowledge is implemented only in phi-3

We have added a discussion on composition in the future outlooks section. It seems likely that the lack of composition is important, and that correlations could be improved by considering it. Mentioned in section 4.3: Composition: one of the major changes in IIT3.0 from previous iterations of the theory is the role of all possible mechanisms (subsets of nodes) to the integration of the system as a whole. To our knowledge, all existing heuristic measures of integrated information are wholistic, always looking at the system as a whole. Future heuristics could take a compositional approach, combining integration values from subsets of measurements, rather than using all measurements at once.”

Page 17 para 3: “testbed for future IIT inspired measure…” -> “testbed for a future IIT inspired measure” - Fixed

Appendix A1 para 1: “Each network were perturbed” -> “Each network was perturbed” - Fixed

“is calculate” -> “are calculated” - Fixed

Appendix A4 para 1: “The average times was recorder…” -> “The average times were recorded…” - Fixed

Appendix A4: Could you report the computational resources used (and if possible, summary of degree of parallelisation implemented / effective resources used for each measure; but likely it is the case that only pyphi implements parallelisation)?

Fixed. Here we did not parallelize PyPhi calculations, opting instead to split at the second highest level possible, with the highest level being parallelizing over networks. As time to compute is highly dependent on how high peak Phi is (especially if peak Phi is over 1), we decided to parallelize over states so as to move as quickly through the heavy networks as possible (2000s for a network, for one state, isn’t unusual in this setup). Edit in A4: “Here we used a 32gb, 16-core (Intel Xeon E5-1660 v4 @ 3.20GHz, 20480 KB), parallelized on the level of states for Φ2.0/2.5/3.0, at the level of partitions (MIP search) for Φ*, SI, and MI1, and none for LZ, S, D1, and D2.”

Table A2: Formatting error at end of caption? “all: measure applied to the whole state transition matrix (STM)” - Fixed

Figure A3: Can add white outlines to each triangle to help visually differentiating close points?

Fixed. Added white outlines to triangles that were competing.

Reviewer 2 Report

See attached.

Author Response

Response to reviewer #2:

We thank the reviewer for their comments and the opportunity to better clarify the overall goal of the paper. We recognize the need to provide more explicit motivation for our study, and specifically, to differentiate our work from the work of Mediano et. al. (2018) Following is our comment to the general issue raised by the reviewer concerning lack of novelty and trivial interpretation:

“This reviewer is happy to see advances made regarding IIT and measures of Φ, especially regarding the application to larger, complex, and real systems. However, in its current form, I am not strongly motivated to see how this study advances understanding of either IIT or measures of Φ. The data analyses and presentation of results are sound. What is lacking is a packaging that sells the project as being unique and/or an advancement/improvement on previous work, especially referenced work. This reviewer wants to see the project succeed, but it needs to motivate its readers to appreciate what is unique about it and what advances are being made.”

First, regarding the motivation for our study. The goal of this study was to evaluate several proposed measures of integrated information, or approximations for integrated information, to see how they relate to integrated information as defined by IIT 3.0. That we are specifically interested in the relation to IIT 3.0 is an important point. It may be that integrated information (broadly construed) has empirical value in the study of consciousness (or complex systems in general), regardless of whether a particular measure is aligned with IIT 3.0. However, here we are not concerned with the empirical value of the proposed measures, but specifically their relation to IIT 3.0, and thus their value for exploring the implications of the theory and their potential to (eventually) be used to test the theory. We have made several edits in both the abstract and introduction to make this motivation more explicit.

Abstract, paragraph 1:

“While these measures and approximations capture intuitions underlying IIT and some have had success in practical applications, it has not been shown that they actually quantify the type of integrated information specified by the latest version of IIT, and thus whether they can be used to test the theory.

Abstract, paragraph 2:

“we stress the importance that measures aimed at being practical alternatives to Φ are at a minimum rigorously tested in an environment where the ground truth can be established”

Introduction,  paragraph 4:

The lack of direct comparisons with Φ3.0 is a gap in the current literature of integrated information methods. If an approximation or heuristic is to be used in an attempt to falsify IIT, then the results are only valid to the extent that the measure accurately estimates Φ3.0 (similarly, for evidence in favour of IIT). It is not possible to validate proposed measures in the networks of interest (due to computational considerations outlined above); however, we can validate measure in smaller systems where Φ3.0 can be calculated directly. We claim that correspondence in smaller systems is a necessary condition for any measure used to evaluate IIT. Therefore, by using deterministic, isolated, discrete networks of binary logic gates of similar type as employed in IIT3.0 [5], this paper aims to evaluate the accuracy relative to Φ3.0 of, (1) approximations that speed up parts of the Φ3.0 calculations, and, (2) heuristic measures of integrated information.

Moreover, we have now framed our conclusion in such a way to directly address this question. We draw the following conclusions:

1.     approximation can be accurate but don’t have a large cost savings – they provide only moderate increase in the size of model systems that can be explored

2.     heuristic have the potential to estimate peak Phi or mean Phi but they are insufficient to evaluate state dependent Phi.

These conclusions are now explicitly stated in the Discussion section,

Discussion, section 4, paragraph 1:

The purpose of the work was to determine which methods, if any, might be used to test the theory.

Discussion, section 4, paragraph 1:

The approximation measures are still computationally intensive and require full knowledge of the systems TPM, meaning they only provide a marginal increase to the size of systems that can be studied.

Discussion, section 4, paragraph 1:

Heuristic measures on the other hand, provide greater reductions in computation and knowledge requirements, and can be applied to much larger systems, but only in a coarser state-independent manner.

Next, regarding the Mediano et al (2018) study. Given the above updates to the motivation, it becomes clear that these two studies are addressing different issues. Our study is specifically interested in heuristics and approximations of Phi3.0, while the Mediano study is more interested in a general notion of integrated information as a measure of complexity, and not with IIT per se. Moreover, in addition to the difference in high-level goals, there are substantial methodological differences between the two studies,

1.     We use perturbation for our observed data, rather than simply observation, making it more “causal” in nature.

2.     We are focused on discrete networks, while Mediano et al (2018)  uses continuous Gaussian variables

3.     We provide a comparison to Phi3.0, Mediano et al (2018)  does not consider this measure (because it is not defined for continuous variables)

4.     Our networks use linear threshold logic, Mediano et al (2018)  uses auto-regressive models

We have added a sentence in the introduction specifically acknowledging the Mediano et al (2018) paper, but noting that their goals are different from ours,

Introduction, paragraph 3:

Notably, [19] compares the behaviour of several heuristic measures developed for time-series data; however, they were interested in the consistency among the methods, rather than a comparison to Φ3.0.

1. Minor issues

We thank the reviewer for finding these mistakes and errors. We have now corrected them and made necessary adjustments. We will only go through them in brief and only add commentary where necessary.

Capitalization of “Integrated information theory” (p. 2) - Fixed

Is IIT an “axiomatic, quantitative approach” as in axiomatic is a quantitative approach; or is it a “axiomatic and quantitative approach”? (p. 2) - Fixed

Although you go into more detail below, it could be helpful to the reader if you explained a bit more how you conceive of the difference between “approximations” and “heuristics” here, and why such a difference is significant for the purposes of the current paper. (p. 2)

Approximations are computational shortcuts within the IIT framework itself, while heuristics are measures capturing central intuitions of IIT. Added the following line in paragraph 3 of introduction: “As Φ3.0 quickly becomes computationally intractable as a function of network size, one approach is to implement approximations; computational shortcuts within the framework of IIT3.0 that reduce computational cost [6]. Another approach is to use heuristic measures that capture central intuitions of IIT such as information differentiation and integration via more tractable methods [10-15].”

Is it that none have validated Φ3.0 or some have “except [15]”? (p. 2)

None, except [15], has validated measures against Φ3.0. Clarified with the following line in paragraph 3 in introduction: “... only [11] has tested a few approximations and heuristics with respect to Φ3.0 in evolved logic-gate based animats.”

I believe parts of the explanation of Φ3.0 in section 2.2 are incorrect, starting “Generally, IIT...”, particularly this part: “how much information a system in a specific state generates above and beyond that of the system’s parts” (p. 4). Consciousness is not identical with how much information is generated, but with an amount of information and its degree of integration.

Rephrased to be more accurate in paragraph 1 of section 2.2: “... the amount of information of a system in a specific state above and beyond the information of the system’s parts.”

In regard to one of the six approximations tested, what does “making an educated guess” mean here? It would be helpful to be more precise. (p. 4) 3

Here we meant that two of the approximations are based on making an a priori estimate of the MC. Clarified in paragraph 1, section 2.3: “... two shortcuts based on making an a priori estimate about the elements included in the MC rather than explicitly testing every candidate subsystem...”

Given the use of linear statistics throughout, it could be helpful to show the results of Φ3.0Peak and Φ3.0Mean. (This certainly does not make or break the paper, but this reviewer is curious, and perhaps other readers would be too.) (p. 16)

If the reviewer here is seeking the comparison of all approximations and heuristics vs. Φ3.0Mean rather than Φ3.0Peak, this has now been added as appendix A5: “...We investigated to what extent replacing 3.0peak with 3.0mean (similarly for measures A-E,G,H) affected the overall results. Analysis and statistical comparisons were performed as in section 2.3.x and 2.4. All of the approximations and heuristics were negligibly affected, suggesting that for small networks of n∈{3,4,5,6} the mean and the the peak state dependent Φ3.0 is interchangeable. See Table A3…”

The phrasing of “...properties to aim for in artificial systems, to produce ‘awareness’” is strange. (p. 17). Do you mean consciousness? Are ‘awareness’ and ‘consciousness’ synonymous or do you mean something different?

Clarified in paragraph 4, section 4.3: “... and properties to aim for in artificial systems to produce “consciousness”.”

Round  2

Reviewer 2 Report

The authors have addressed my concerns. I look forward to seeing this paper published.

Author Response

Thank you for your detailed review and comments. We are glad that your concerns has been answered satisfactorily.